# A Copper Foil Electromagnetic Coupler and Its Wireless Power Transfer System without Compensation

**Xueying Wu [1,*] and Mingxuan Mao [2]**

1 School of Electrical Engineering, Chongqing University of Science & Technology, No.20, East Road, University Town, Shapingba District, Chongqing 401331, China

2 School of Electrical Engineering, Chongqing University, 174 Shazheng Street, Shapingba District, Chongqing 400044, China; mx_m@cqu.edu.cn

* Correspondence: 2020021@cqust.edu.cn

**Abstract:** This paper proposes a copper foil electromagnetic coupler integrating inductance and capacitance and its wireless power transfer (WPT) system without additional compensation structure. Firstly, the equivalent circuit model of the integrated electromagnetic coupler is established, and the circuit model is simplified based on the circuit theory and mutual inductance coupling theory. The self-compensating characteristics of the coupler are utilized to analyze and design the relation between electrical parameters of the system, and the basic conditions of full resonance working of the system are given. The system's performance is verified by simulation.

**Keywords:** wireless power transfer; capacitive power transfer; copper foil electromagnetic coupler

## 1. Introduction

Wireless power transfer (WPT) technology can effectively solve the problem of unsafe and inflexible access in traditional charging mode [1–4]. For inductive power transfer (IPT) systems, four classical compensation structures are usually used to compensate reactive power to improve the system performance [5]. However, there are also many problems in the four compensation structures. The compensation capacitance of the receiver in P/S and P/P structure is related to the load [6]; the inverter output voltage of S/S structure is very sensitive to the load change [7]; the output voltage of the S/P compensation structure at the gain intersection is approximately inversely proportional to the coupling coefficient [8]. For capacitive power transfer (CPT) system, a very large inductance is generally used to compensate reactive power in series for long-distance (cm level) transmission applications, which increases the cost and weight of the system [9]. In addition, for the application of high-power wireless charging, the single inductance compensation structure will cause the coupler to bear high voltage stress.

Aiming at the above problems of CPT and IPT systems, various mix-compensation structures, such as LCL, LCC, CLC, and LCLC, have been put forward and improved in succession [10–13]. However, the Litz wire coil and high-order compensation network lead to the increase in system volume and cost, and greatly reduce the power density of the system, which seriously restricts the application and promotion of WPT technology.

In order to further improve the performance of WPT system, a lot of research has been conducted on electromagnetic couplers. In recent years, much attention has been paid to hybrid inductive and capacitive electromagnetic couplers, which were first proposed in [14]; such a coupler raises the transmission efficiency and improves the misalignment ability of the system. On the basis of [14], the relationship between IPT and CPT was analyzed in detail in [15], and the hybrid WPT system was further refined to realize high efficiency and high power transmission. Reference [16] utilized another distinctive hybrid inductive and capacitive electromagnetic coupler to transfer electric power crossing metal obstacles. In [17], a new hybrid electromagnetic coupler was put forward, in which a

circular coupling coil is embedded in a square metal frame to save power transfer space and improve transfer distance.

In most cases, in order to improve the transfer distance, output power and efficiency of the WPT system, a corresponding high-order compensation network must be added for the coupler, whether it is a single magnetic coupler, a single electric coupler, or a hybrid electromagnetic coupler. Generally, the magnetic coupler is usually made of Litz wire, while the electric coupler generally consists of an aluminum or copper metal plate, which increases the system volume, weight and cost. So the power density of the system will be greatly reduced, which seriously restricts the application and promotion of WPT technology.

In view of the above problems, this paper starts to study the inherent characteristics of coupler. Aiming at simplifying the system structure, reducing the system cost and improving the system power density, a copper foil electromagnetic coupler and the optimization design method for its WPT system are proposed. In order to overcome the influence of skin effect on the system and reduce the ohmic loss of the coupler, the planar rectangular spiral coil made of copper foil is used as the electromagnetic pole. The four electromagnetic poles are stacked to form the electromagnetic coupler. They possess a certain self-inductance, and cross coupling will occur between the four electromagnetic poles, forming mutual inductances and cross-coupling capacitances. The equivalent circuit of the coupler is simplified, and the AC impedance model of the system is established. The self-compensating equation is derived and the working condition of zero phase angle (ZPA) is given. The proposed coupler and WPT system are verified by simulation and experimental results.

## 2. Inductance and Capacitance Integrated Electromagnetic Coupler

The proposed integrated electromagnetic coupler is shown in Figure 1, which is different from the coil made of Litz wire in the IPT system and the electrode made of metal plate in the CPT system. In this paper, the electromagnetic pole is a spiral wound with square copper foil with a certain width, which can not only ensure the self-inductance and mutual inductance of the electromagnetic pole, but also increase the cross coupling capacitance between the electromagnetic poles. By making full use of the parasitic capacitance characteristics of the coil ignored in previous studies, the power transmission quality can be improved. In Figure 1b, $D_{out}$, $D_{in}$, $w$, and $s$ are the outer diameter of the electromagnetic pole, the inner diameter of the electromagnetic pole, the width of the copper foil, and the distance between two turns of copper foil, respectively. The four electromagnetic poles of the coupler are stacked to cause cross coupling with each other. The mutual inductance and mutual capacitance among the four electromagnetic poles must be considered when analyzing the characteristics of the coupler. Therefore, the structure in Figure 2a can be equivalent to the circuit model shown in Figure 2b. $L_i$ is the self-inductance of the electromagnetic pole, $M_{ij}$ is the mutual inductance of $P_i$ and $P_j$, $C_{ij}$ is the cross-coupling capacitance of $P_i$ and $P_j$, where $i, j = 1, 2, 3, 4$.

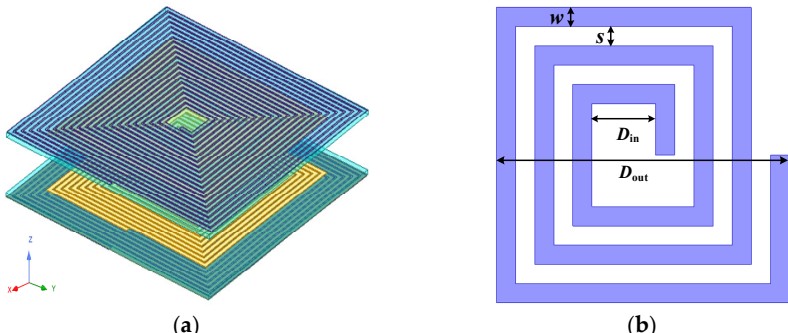

| (a) | (b) |

**Figure 1.** Basic structure of integrated electromagnetic coupler. (**a**) 3D Graph of integrated electromagnetic coupler. (**b**) Top view of plane rectangular spiral electromagnetic pole.

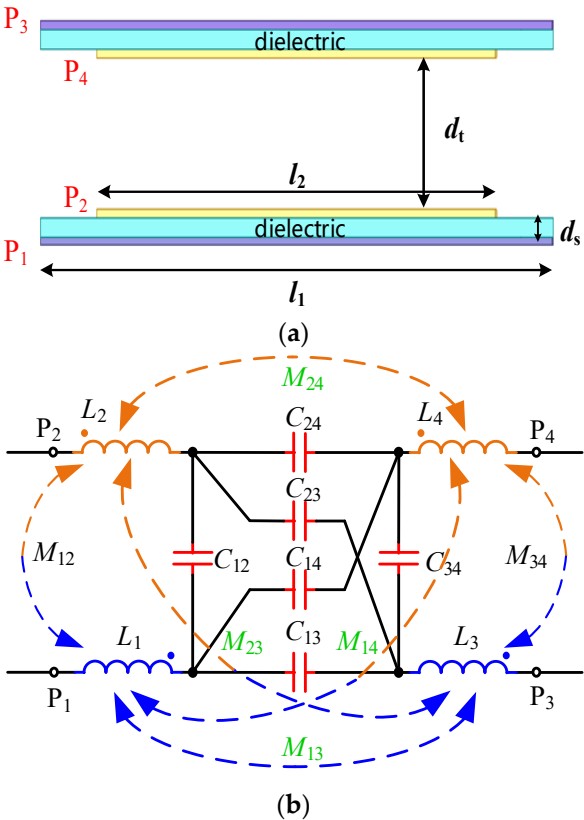

**Figure 2.** Coupling structure and equivalent model. (**a**) Front view of integrated electromagnetic coupler. (**b**) Equivalent model of integrated electromagnetic coupler.

## 3. System Circuit Modelling and Self-Compensating Principle

### 3.1. System Circuit Model and Its Equivalent Simplification

In order to simplify the WPT system structure, no other compensation components are added to the system, and the capacitance and inductance characteristics of the copper foil electromagnetic coupler are utilized to realize self-compensation. The non-compensating structure is presented in Figure 3, where the inner electromagnetic pole $P_2$ of the coupler is connected to the high potential terminal of the inverter output, and the outer electromagnetic pole $P_1$ is connected to the low potential terminal, so as to reduce the leakage electric field. For convenience, the fundamental harmonics approximation (FHA) method is used to analyze the circuit characteristics. The power losses in the coupler are also neglected. For the circuit model of the coupler shown in Figure 2, according to reference [18], the six capacitance cross coupling model can be equivalent to a three capacitance $\pi$ model, so the system circuit model can be equivalent to the circuit model shown in Figure 4. Combined with the multi inductance coupling theory, the inductance components $L_1$–$L_4$ can be decoupled, and the circuit shown in Figure 4 can be simplified to the circuit structure shown in Figure 5.

The parameters in Figures 3–5 have the following equivalent transformation relations:

$$\begin{cases} C_M = \frac{C_{13}C_{24}-C_{14}C_{23}}{C_{13}+C_{14}+C_{23}+C_{24}} \\ C_1 = C_{12} - C_M + \frac{(C_{13}+C_{14})(C_{23}+C_{24})}{C_{13}+C_{14}+C_{23}+C_{24}} \\ C_2 = C_{34} - C_M + \frac{(C_{13}+C_{23})(C_{14}+C_{24})}{C_{13}+C_{14}+C_{23}+C_{24}} \end{cases} \tag{1}$$

$$\begin{cases} C_A = C_1 + C_M + \frac{C_1 \cdot C_M}{C_2} \\ C_B = C_1 + C_2 + \frac{C_1 \cdot C_2}{C_M} \\ C_C = C_2 + C_M + \frac{C_2 \cdot C_M}{C_1} \end{cases} \tag{2}$$

$$\begin{cases} L_{\mathrm{M}} = -M_{13} - M_{14} - M_{23} - M_{24} \\ L_{1\mathrm{M}} = L_1 + M_{12} + M_{13} + M_{14} \\ L_{2\mathrm{M}} = L_2 + M_{21} + M_{23} + M_{24} \\ L_{3\mathrm{M}} = L_3 + M_{31} + M_{32} + M_{34} \\ L_{4\mathrm{M}} = L_4 + M_{41} + M_{42} + M_{43} \end{cases} \tag{3}$$

The electric field coupling coefficient and magnetic field coupling coefficient are defined as

$$k_{\mathrm{CPT}} = \frac{C_{\mathrm{M}}}{\sqrt{(C_1 + C_{\mathrm{M}})(C_2 + C_{\mathrm{M}})}} \tag{4}$$

$$k_{\mathrm{IPT}} = \frac{L_{\mathrm{M}}}{\sqrt{(L_{1\mathrm{M}} + L_{2\mathrm{M}})(L_{3\mathrm{M}} + L_{4\mathrm{M}})}} \tag{5}$$

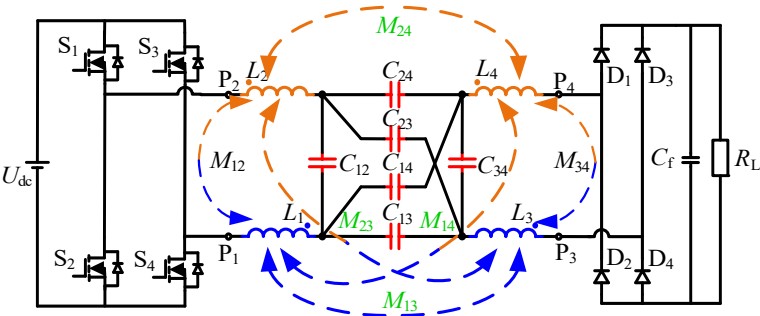

**Figure 3.** Circuit topology of WPT system without compensation.

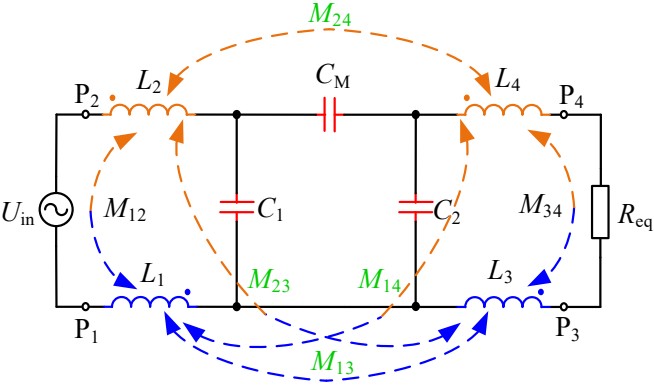

**Figure 4.** System π-type model.

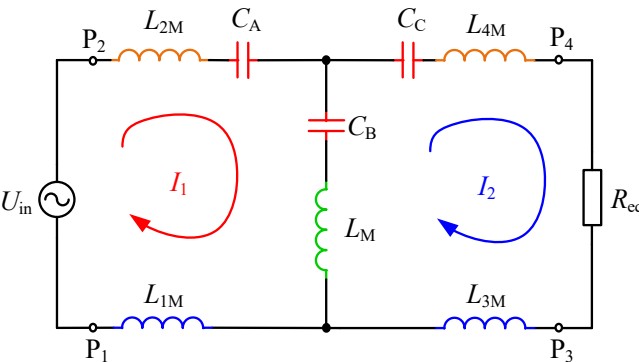

**Figure 5.** System T-type decoupled model.

### 3.2. Circuit Full Resonance and Parameter Self-Compensating Relation

According to the simplified circuit topology shown in Figure 5, the circuit is divided into two loops. The loop circuits are $I_1$ and $I_2$, respectively. The current clockwise direction is specified as the positive direction. The loop equation is written based on Kirchhoff's voltage law (KVL):

$$\begin{bmatrix} a_{11} & a_{12} \\ a_{21} & a_{22} \end{bmatrix} \cdot \begin{bmatrix} \dot{I}_1 \\ \dot{I}_2 \end{bmatrix} = \begin{bmatrix} \dot{U}_{\text{in}} \\ 0 \end{bmatrix} \tag{6}$$

where

$$\begin{cases} a_{11} = j\omega(L_{1M} + L_{2M} + L_M) + \frac{1}{j\omega C_A} + \frac{1}{j\omega C_B} \\ a_{12} = -\left(j\omega L_M + \frac{1}{j\omega C_B}\right) \\ a_{21} = -\left(j\omega L_M + \frac{1}{j\omega C_B}\right) \\ a_{22} = j\omega(L_{3M} + L_{4M} + L_M) + \frac{1}{j\omega C_B} + \frac{1}{j\omega C_C} + R_{\text{eq}} \end{cases} \tag{7}$$

where $\omega = 2\pi f$, and $f$ is the system operating frequency. The loop currents can be calculated as

$$\begin{cases} \dot{I}_1 = \frac{a_{22}}{a_{11}a_{22} - a_{12}a_{21}} \cdot \dot{U}_{\text{in}} \\ \dot{I}_2 = \frac{-a_{21}}{a_{11}a_{22} - a_{12}a_{21}} \cdot \dot{U}_{\text{in}} \end{cases} \tag{8}$$

$\dot{I}_1$ and $\dot{I}_2$ can also be regarded as the system input current $\dot{I}_{\text{in}}$ and output current $\dot{I}_{\text{out}}$. So the system input impedance can be expressed as

$$Z_{\text{in}} = \frac{\dot{U}_{\text{in}}}{\dot{I}_1} = \frac{a_{11}a_{22} - a_{12}a_{21}}{a_{22}} \tag{9}$$

In practical application, the four stacked copper foil coupler is generally designed as a symmetrical structure, i.e., the dimensions of electromagnetic pole $P_1$ and $P_3$ are identical, and the dimensions of electromagnetic pole $P_2$ and $P_4$ are also identical. If the system is required to work in ZPA state, i.e., the imaginary part of input impedance $\text{Im}(Z_{\text{in}}) = 0$, the following equation must be satisfied:

$$\omega^2 \frac{C_A C_B}{C_A + C_B}(L_{1M} + L_{2M} + L_M) = 1 \tag{10}$$

or

$$\frac{2\omega^2 L_M C_A C_B + 2\omega^2 C_A(L_{1M} + L_{2M})(C_A + C_B)}{2\omega^4 C_A^2 C_B(L_{1M} + L_{2M})(L_{1M} + L_{2M} + 2L_M) + \omega^2 C_A^2 C_B R_{\text{eq}}^2 + 2C_A + C_B} = 1 \tag{11}$$

*(1)* Equation (10) is satisfied.

The system operating angular frequency can be calculated as

$$\omega = \sqrt{\frac{C_A + C_B}{(L_{1M} + L_{2M} + L_M)C_A C_B}} \tag{12}$$

The input current and output current can be expressed as

$$\begin{cases} \dot{I}_{\text{in}} = \frac{(L_M + L_{1M} + L_{2M})(C_A + C_B)C_A C_B R_{\text{eq}}}{((L_{1M} + L_{2M})C_A - L_M C_B)^2} \dot{U}_{\text{in}} \\ \dot{I}_{\text{out}} = -j \frac{\sqrt{(L_M + L_{1M} + L_{2M})(C_A + C_B)C_A C_B}}{(L_{1M} + L_{2M})C_A - L_M C_B} \dot{U}_{\text{in}} \end{cases} \tag{13}$$

So the output voltage of the system is further obtained as follows:

$$\dot{U}_{\text{out}} = \dot{I}_{\text{out}} R_{\text{eq}} = -j\dot{U}_{\text{in}} R_{\text{eq}} \frac{\sqrt{(L_M + L_{1M} + L_{2M})(C_A + C_B)C_A C_B}}{(L_{1M} + L_{2M})C_A - L_M C_B} \tag{14}$$

Ignoring the parasitic resistance of the coupler, the output active power of the WPT system equals the input active power since the coupler is lossless. The input power factor can be expressed as

$$\lambda = \left| \frac{\mathrm{Re}\left( \dot{U}_{\mathrm{in}} \dot{I}_{\mathrm{in}} \right)}{\dot{U}_{\mathrm{in}} \dot{I}_{\mathrm{in}}} \right| = \left| \frac{\dot{U}_{\mathrm{out}} \dot{I}_{\mathrm{out}}}{\dot{U}_{\mathrm{in}} \dot{I}_{\mathrm{in}}} \right| = 1 \tag{15}$$

*(2)　Equation (11) is satisfied.*

The system operating angular frequency can be calculated as

$$\begin{cases} \omega_1 = \sqrt{\dfrac{-B + \sqrt{B^2 - 4AC}}{2A}} \\ \omega_2 = \sqrt{\dfrac{-B - \sqrt{B^2 - 4AC}}{2A}} \end{cases} \tag{16}$$

where

$$\begin{aligned} A &= C_{\mathrm{A}}^2 C_{\mathrm{B}} (L_{1\mathrm{M}} + L_{2\mathrm{M}})(L_{1\mathrm{M}} + L_{2\mathrm{M}} + L_{\mathrm{M}}) \\ B &= C_{\mathrm{A}}^2 C_{\mathrm{B}} R_{\mathrm{eq}}^2 - 2C_{\mathrm{A}} C_{\mathrm{B}} L_{\mathrm{M}} - 2C_{\mathrm{A}}(L_{1\mathrm{M}} + L_{2\mathrm{M}})(C_{\mathrm{A}} + C_{\mathrm{B}}) \\ C &= 2C_{\mathrm{A}} + C_{\mathrm{B}} \end{aligned} \tag{17}$$

Combining (9), the input impedance $Z_{\mathrm{in}}$ can be simplified as $R_{\mathrm{eq}}$. Similarly, the input current, output current, and output voltage of the system can be expressed as follows:

$$\dot{I}_{\mathrm{in}} = \frac{\dot{U}_{\mathrm{in}}}{R_{\mathrm{eq}}} \tag{18}$$

$$\dot{I}_{\mathrm{out}} = -\frac{\dot{U}_{\mathrm{in}}}{R_{\mathrm{eq}}} \cdot \frac{\omega^2 C_{\mathrm{A}}(L_{1\mathrm{M}} + L_{2\mathrm{M}}) - 1 + j\omega C_{\mathrm{A}} R_{\mathrm{eq}}}{\omega^2 C_{\mathrm{A}}(L_{1\mathrm{M}} + L_{2\mathrm{M}}) - 1 - j\omega C_{\mathrm{A}} R_{\mathrm{eq}}} \tag{19}$$

$$\dot{U}_{\mathrm{out}} = \dot{I}_{\mathrm{out}} R_{\mathrm{eq}} = -\dot{U}_{\mathrm{in}} \cdot \frac{\omega^2 C_{\mathrm{A}}(L_{1\mathrm{M}} + L_{2\mathrm{M}}) - 1 + j\omega C_{\mathrm{A}} R_{\mathrm{eq}}}{\omega^2 C_{\mathrm{A}}(L_{1\mathrm{M}} + L_{2\mathrm{M}}) - 1 - j\omega C_{\mathrm{A}} R_{\mathrm{eq}}} \tag{20}$$

The input power factor can be expressed as

$$\begin{aligned} \lambda &= \left| \frac{\mathrm{Re}\left( \dot{U}_{\mathrm{in}} \dot{I}_{\mathrm{in}} \right)}{\dot{U}_{\mathrm{in}} \dot{I}_{\mathrm{in}}} \right| = \left| \frac{\dot{U}_{\mathrm{out}} \dot{I}_{\mathrm{out}}}{\dot{U}_{\mathrm{in}} \dot{I}_{\mathrm{in}}} \right| \\ &= \left| \frac{\left( \omega^2 C_{\mathrm{A}}(L_{1\mathrm{M}} + L_{2\mathrm{M}}) - 1 + j\omega C_{\mathrm{A}} R_{\mathrm{eq}} \right)^2}{\left( \omega^2 C_{\mathrm{A}}(L_{1\mathrm{M}} + L_{2\mathrm{M}}) - 1 - j\omega C_{\mathrm{A}} R_{\mathrm{eq}} \right)^2} \right| = 1 \end{aligned} \tag{21}$$

## 4. Analysis and Design of Circuit Parameters and Electromagnetic Coupler Parameters

*4.1. Selection for System Working Frequency Considering the Influence of Inner Electromagnetic Poles' Copper foil Turns*

In order to improve the system performance, the finite element analysis software Maxwell is used to analyze the characteristics of coupler with different dimensions and optimize the spatial structure of the coupler. Polymethyl methacrylate (PMMA, the relative permittivity is 3.4) is used as the dielectric between the electromagnetic poles on the same side, and the parameters of the coupler shown in Table 1 are used as invariants.

**Table 1.** Geometric parameters of electromagnetic coupler.

| Parameter | $N_{\mathrm{out}}$ | $s_{\mathrm{out}}$ | $w_{\mathrm{out}}$ | $s_{\mathrm{in}}$ | $w_{\mathrm{in}}$ | $d_{\mathrm{s}}$ | $d_{\mathrm{t}}$ |
|---|---|---|---|---|---|---|---|
| Value | 15 | 5 mm | 10 mm | 5 mm | 10 mm | 10 mm | 60 mm |

For the first ZPA condition, according to the analysis in Section 3.2, we can obtain the variation of the system working frequency *f* with $C_{\mathrm{A}}$, $C_{\mathrm{B}}$, $L_{\mathrm{M}}$, $L_{1\mathrm{M}}$ and $L_{2\mathrm{M}}$. The variation of the above capacitances and inductances with $N_{\mathrm{in}}$ can be simulated by Maxwell, as shown in Figure 6a,b. Further combining (10), the variation of the input impedance $Z_{\mathrm{in}}$ with $N_{\mathrm{in}}$

and $R_{eq}$ is obtained, as shown in Figure 7. It can be seen that $N_{in}$ has little effect on $Z_{in}$ which is mainly decided by $R_{eq}$. When the value of $R_{eq}$ is small (10–100 $\Omega$), the input impedance is very high, so the pick-up power of the load will be greatly reduced in this case. When the value of $R_{eq}$ is large ($10^3$–$10^4$ $\Omega$), the input impedance varies from 10 $\Omega$ to 100 $\Omega$, but the load pick-up current is very small. In addition, we can also analyze the influence of the other parameters in (9) on $Z_{in}$ through the same way. $Z_{in}$ is less effected by the other parameters, which can also be verified by Maxwell simulation.

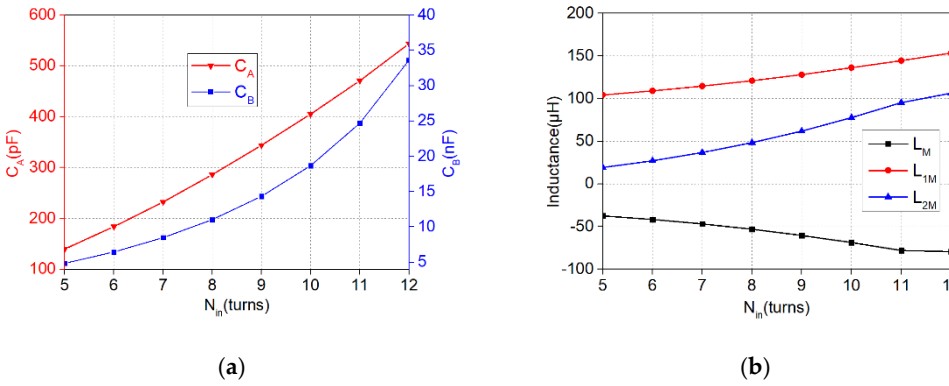

(**a**)                                                                                               (**b**)

**Figure 6.** Variation of the equivalent cross-coupling capacitances and mutual inductances. (**a**) Equivalent cross-coupling capacitances. (**b**) Equivalent mutual inductances.

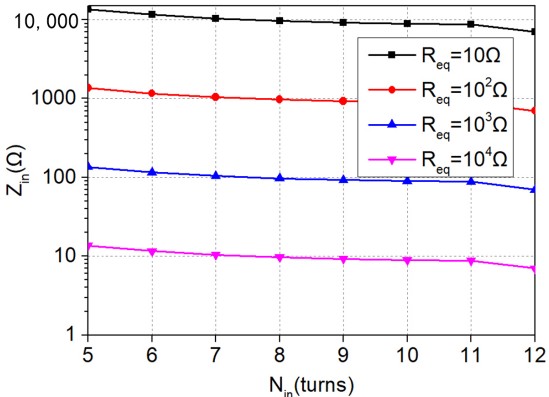

**Figure 7.** Variation of the input impedance.

Based on the above analysis, the first ZPA working mode is only suitable for applications where the equivalent load resistance is large and the load current is small. In most practical applications, the equivalent load resistance is generally within 100 $\Omega$. Next, this paper will focus on the analysis and optimization design of the electromagnetic coupler in the second ZPA condition.

For the second ZPA condition, the system input impedance is equal to the equivalent load resistance ($Z_{in}$ = $R_{eq}$), and the voltage and current of the load are more suitable for most applications. Similarly, the parameters of the coupler shown in Table 1 are taken as invariants, and the changes of the above two frequencies with $N_{in}$ and $R_{eq}$ are obtained, as shown in Figure 8. It is concluded that the system resonance frequencies in the second ZPA condition are mainly related to $N_{in}$ and less affected by $R_{eq}$.

In order to choose a more suitable system frequency, this paper will analyze the voltage to ground of each electromagnetic pole under the above two resonant frequencies. According to Figure 3, the electromagnetic pole $P_1$ is connected to the negative pole of DC input and $P_2$ is connected to the high potential terminal of inverter output, so the voltage on $P_1$ is zero and the voltage on $P_2$ is $U_{in}$. The voltage on $P_3$ and $P_4$ will be mainly considered for comparative analysis. With different values of $N_{in}$, the resonance

frequency of the coupler is quite different, which will cause the voltages to ground of the four electromagnetic poles distinct. When the system operating frequency is equal to $f_1$ and $f_2$, the variation of voltage to ground of $P_3$ and $P_4$ with $N_{in}$ can be obtained, as shown in Figure 9. It is obvious that the $U_{P3}$ and $U_{P4}$ obtained by using $f_2$ as the system working frequency are much higher than those obtained by using $f_1$ as the system working frequency. Therefore, choosing $f_1$ as the system working frequency will effectively reduce the fringing electric field of the coupler and improve the safety performance of the system. In the following paper, we will mainly analyze and design the WPT system in the case of $f = f_1$.

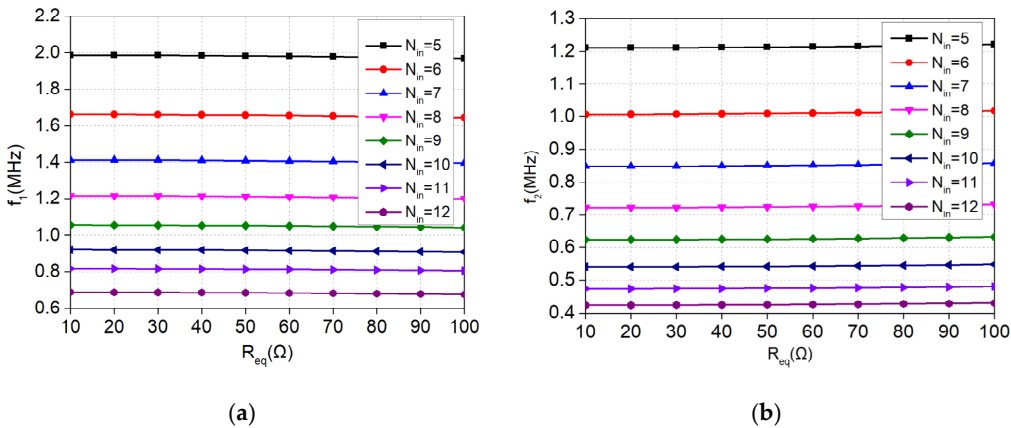

**Figure 8.** The variation of system operating frequency with $N_{in}$ and $R_{eq}$. (**a**) The variation of $f_1$ with $N_{in}$ and $R_{eq}$. (**b**) The variation of $f_2$ with $N_{in}$ and $R_{eq}$.

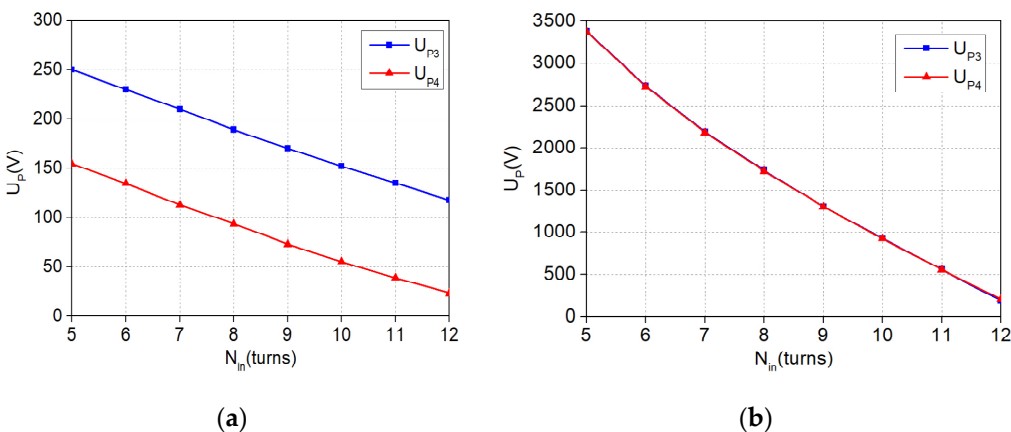

**Figure 9.** Variation curve of the voltage to ground of $P_3$ and $P_4$ with $N_{in}$. (**a**) $f = f_1$. (**b**) $f = f_2$.

### 4.2. Influence of Misalignment on Coupling Coefficient

The proposed electromagnetic poles are square, so the variation of the mutual inductance and cross-coupling capacitance caused by displacement misalignment in X direction and Y direction is basically consistent. Therefore, the varying differences in X direction and Y direction of magnetic field coupling coefficient $k_{IPT}$ and electric field coupling coefficient $k_{CPT}$ can be negligible when there is displacement misalignment. Figure 10 shows the variation of $k_{IPT}$ and $k_{CPT}$ at X or Y misalignment conditions. It can be seen that the sign of $k_{IPT}$ and $k_{CPT}$ will change with the misalignment increasing. $k_{IPT}$ will equal zero when the misalignment increases to 294.3 mm, while $k_{CPT}$ will equal zero when the misalignment equals 275.8 mm, which means the power cannot be transferred at this point.

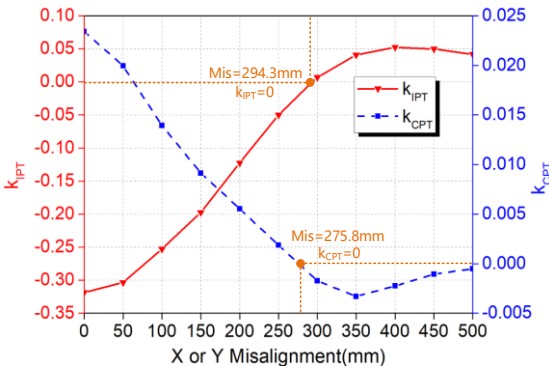

**Figure 10.** The variation of $k_{\text{IPT}}$ and $k_{\text{CPT}}$ at X or Y misalignment conditions.

## 5. Parameters Determination and Simulation Verification

　　Based on the circuit shown in Figures 3 and 4 and the geometric parameters of electromagnetic coupler in Table 2, the circuit parameters can be calculated as Table 3, and then the system simulation model is built on MATLAB/Simulink. The relationship between the input impedance characteristics and the system working frequency is obtained as shown in Figure 11. Generally, $f_1$ and $f_2$ are selected as the working frequency of the system, which is more in line with the equivalent load resistance in practical application. However, compared with the frequency of $f_2$, the voltage on electromagnetic pole is greatly reduced under the frequency of $f_1$, as can be seen from Figure 9. Therefore, $f_1$ will be selected as the working frequency of the system in the following simulation. Figure 12 shows the output voltage and current waveform the inverter. The voltage and current are in the same phase, and the system realizes ZPA operation with a unity power factor in the input. The measurement shows that the input current amplitude of the system is 2.484 A. Because $U_{\text{in}} = U_{\text{dc}} \times 4/\pi$ and $R_{\text{eq}} = R_{\text{L}} \times 8/\pi^2$, the input current $I_{\text{inv}}$ can be calculated as $I_{\text{in}} = U_{\text{in}}/R_{\text{eq}} = 2.5\text{A}$, which is consistent with the simulation results. Because the internal resistance loss of the electromagnetic pole is not considered in the simulation, the output power is basically consistent with the input power.

**Table 2.** Geometric parameters of electromagnetic coupler.

| Parameter | $N_{\text{out}}$ | $s_{\text{out}}$ | $w_{\text{out}}$ | $s_{\text{in}}$ | $w_{\text{in}}$ | $d_{\text{s}}$ | $d_{\text{t}}$ |
|-----------|------------------|------------------|------------------|-----------------|-----------------|----------------|----------------|
| value | 15 | 5 mm | 10 mm | 5 mm | 10 mm | 10 mm | 60 mm |

**Table 3.** Circuit simulation parameters.

| Parameter | Value | Parameter | Value |
|-----------|-------|-----------|-------|
| $f$ | 1.056 MHz | $L_2$ | 19.517 μH |
| $U_{\text{dc}}(U_{\text{in}})$ | 78.54 V (100 V) | $L_3$ | 64.407 μH |
| $R_{\text{L}}(R_{\text{eq}})$ | 49.348 Ω (40 Ω) | $L_4$ | 19.432 μH |
| $C_{12}$ | 324.98 pF | $M_{12}$ | 22.731 μH |
| $C_{13}$ | 30.376 pF | $M_{13}$ | 28.369 μH |
| $C_{14}$ | 3.0998 pF | $M_{14}$ | 12.367 μH |
| $C_{23}$ | 3.1395 pF | $M_{23}$ | 12.366 μH |
| $C_{24}$ | 13.198 pF | $M_{24}$ | 7.2848 μH |
| $C_{34}$ | 324.14 pF | $M_{34}$ | 22.722 μH |
| $L_1$ | 64.622 μH | | |

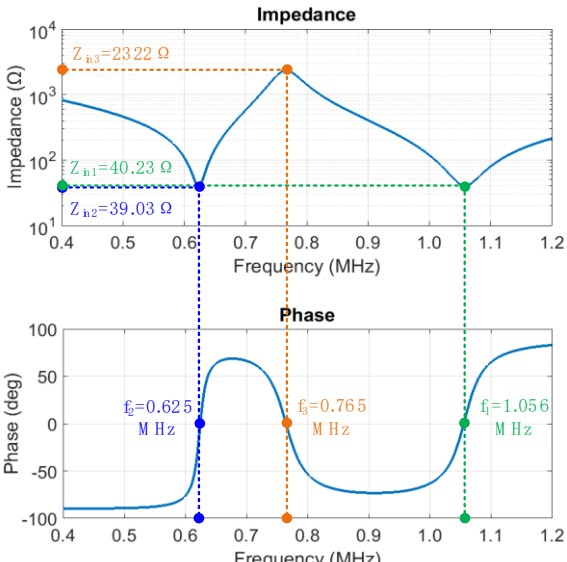

**Figure 11.** Relationship between input impedance and working frequency.

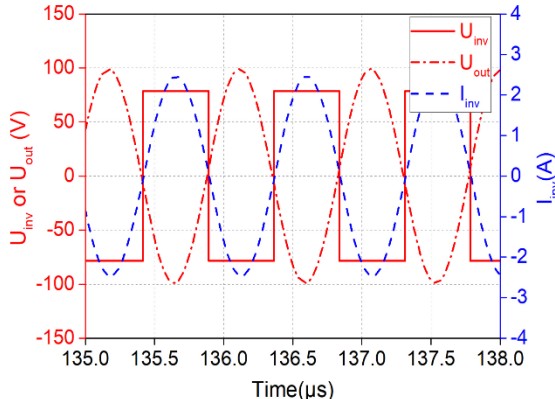

**Figure 12.** Waveform of inverter output voltage and current.

## 6. Conclusions

This paper proposes a copper foil electromagnetic coupler possessing a self-compensating characteristic, which integrates inductance and capacitance. Its wireless power transfer (WPT) system without additional compensation structure is further presented. This coupler and its WPT system have the following advantages:

(1) Low cost, light weight, simple structure, and high power density;
(2) No additional compensation components and little skin effect, so as to improve system efficiency;
(3) High power factor and ZVS condition.

**Author Contributions:** This paper is mainly written and edited by X.W. and reviewed by M.M. All authors have read and agreed to the published version of the manuscript.

**Funding:** This research was funded by initial Scientific Research Fund of Young Teachers in Chongqing University of Science and Technology (182003004).

**Institutional Review Board Statement:** Not applicable.

**Informed Consent Statement:** Not applicable.

**Conflicts of Interest:** The authors declare no conflict of interest. The funders had no role in the design of the study; in the collection, analyses, or interpretation of data; in the writing of the manuscript, or in the decision to publish the results.

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
