# Peer review of "A Copper Foil Electromagnetic Coupler and Its Wireless Power Transfer System without Compensation"

_wevj, doi:10.3390/wevj12040191_

Round 1

Reviewer 1 Report

The main contribution of this manuscript is using the parasitic capacitance of the coils for inductive wireless power transfer applications.

  1. It is difficult to prove the proposed idea using circuit simulation. some current or electromagnetic fields can flow through the parasitic capacitors, so 3D simulation of the proposed coils is required and using the results parasitic capacitance values are extracted.
  2. power transfer efficiency and power factor should be presented. In spite of zero coil resistance, the efficiency does not seem to be high. 
  3. To understand the simulated input and output results, the whole coupler circuit topology should be presented in detail.

Reviewer 2 Report

Q1 The introduction is too short. The state of the art is rather poorly presented.

Q2 Authors select the f3 as working frequency, what specification can be used this frequency in WPT?

Q3 There is no comparison table.

Q4 The conversion efficiency of input and output is not provided. Please simulate the input and output of the AC waveform on the same figure.

Q5 In Fig.6, the unit is not right on the x-axis.

Q6 Plotting the simulated coupling coefficient against the degree of misalignment

Round 2

Reviewer 1 Report

It seems that the review opinions raised in the first review were revised and improved.

Reviewer 2 Report

Highly improved in this version.